# Elevated Platelet-to-Lymphocyte Ratio and Neutrophil-to-Lymphocyte Ratio after First Cycle of Chemotherapy and Better Survival in Esophageal Cancer Patients Receiving Concurrent Chemoradiotherapy

**Ruo-Han Tseng [1],[†], Kuan-Ming Lai [1],[†], Chien-Yu Tsai [1] and Sheng-Lei Yan [2],[3],***

[1]  Division of Hematology and Oncology, Department of Internal Medicine, Changhua Christian Hospital, Changhua 500, Taiwan
[2]  Division of Gastroenterology, Department of Internal Medicine, Chang-Bing Show Chwan Memorial Hospital, Changhua 505, Taiwan
[3]  Department of Food Science and Biotechnology, College of Biotechnology and Bioresources, Dayeh University, Changhua 500, Taiwan
*   Correspondence: dm467@cbshow.org.tw; Tel.: +886-4-7813888 (ext. 70107); Fax: +886-4-7073226
†   This authors contributed equally to this work.

**Abstract:** *Background:* Prognostic factors for poor survival have been proposed in esophageal squamous cell carcinoma (SCC) patients receiving concurrent chemoradiotherapy (CCRT). We conducted a retrospective study on hematological profile after first cycle of chemotherapy for esophageal SCC patients receiving CCRT. *Methods*: From January 2008 to December 2017, a total of 420 patients with esophageal SCC were enrolled. All included patients had undergone CCRT. Complete blood count, differential count, NLR, and PLR before chemotherapy (CHT) and after first cycle of CHT were obtained. Univariate and multivariate Cox regression analyses were used to assess the association between survival and patient, disease, and treatment characteristics. *Results:* On univariate analysis, significant factors for overall survival (OS) and disease specific survival (DSS) included ECOG performance status, clinical staging, operation, cisplatin dose, prechemotherapy NLR and PLR, and elevated postchemotherapy NLR. On multivariate analysis, ECOG performance status 0–I, Clinical staging I–II, Operation, cisplatin dose >150 mg/m$^2$, prechemotherapy PLR <375, and postchemotherapy platelet count ≥150 × 10$^9$/L were independent factors for predicting better OS. Independent factors for predicting better DSS included ECOG performance status 0–I, Clinical staging I–II, Operation, cisplatin dose >150 mg/m$^2$, and prechemotherapy PLR <375. *Conclusions*: Our study showed that low levels of prechemotherapy PLR and NLR were associated with better OS and DSS. Elevated platelet count and NLR after first cycle of CHT were associated with better OS. Elevated PLR and NLR after first cycle of CHT were associated with better DSS.

**Keywords:** platelet-to-lymphocyte ratio; neutrophil-to-lymphocyte ratio; survival; esophageal cancer; concurrent chemoradiotherapy

## 1. Introduction

Esophageal cancer is the eighth most common cancer globally, with squamous cell carcinoma (SCC) accounting for 90% of cases [1]. The treatment of esophageal cancer included esophagectomy, neoadjuvant chemotherapy (CHT), radiotherapy (RT), chemoradiotherapy, or combination modality. Concurrent chemoradiotherapy (CCRT) has been proved to be superior to RT alone in treatment of locally advanced esophageal SCC [2,3]. Preoperative CCRT provided more survival benefits to patients with locally advanced esophageal SCC than that of surgery alone [4,5]. In esophageal cancer patients receiving CCRT, advanced stage, poor performance status and poor response to CCRT were independent prognostic

factors for poor survival [6]. In a meta-analysis by Zhao QT et al. [7], elevated pretreatment platelet-to- lymphocyte ratio (PLR) significantly predicted poor overall survival (OS), disease free survival (DFS) and cancer-specific survival (CSS) for esophageal cancer patients. Another meta-analysis also showed that high pretreatment PLR was significantly predictive of poor OS, especially in a subgroup of patients who received surgery without pretreatment CCRT [8]. In a meta-analysis of 20 studies and 6457 patients with esophageal cancer [9], elevated preoperative neutrophil-to- lymphocyte ratio (NLR) was shown to be a predictor for poor OS, DFS, and progression free survival (PFS). An update meta-analysis by Binfeng Li et al., also showed that elevated pretreatment NLR might predict poor OS, CSS, PFS and DFS for patients with esophageal cancer [10]. However, little is known about the association of postchemotherapy PLR and NLR levels and treatment outcome. Another study by Tankel J et al., showed that perioperative change in PLR $\geq$ 43.4 was associated with poor overall survival in patients with esophageal adenocarcinoma [10]. Therefore, we conducted a retrospective study with an emphasis on hematological profile after first cycle of chemotherapy to find out prognostic factors for esophageal squamous cell carcinoma (SCC) patients receiving CCRT.

## 2. Methods

This was a single-institution, retrospective, cohort study. From January 2008 to December 2017, patients diagnosed of having esophageal SCC at Changhua Christian Hospital were enrolled. The definitive therapies for these patients were administered according to the guideline of our institution and the discussion among multidisciplinary medical team members. All included patients had undergone CCRT. The study was reviewed and approved by the institutional review board (IRB) of Changhua Christian Hospital (IRB No. 190123). The IRB approved a waiver for informed consent. Basic demographic data were recorded, including age, gender, Eastern Cooperative Oncology Group (ECOG) performance status, and initial presentation of the primary tumor (e.g., tumor site, tumor-node-metastasis (TNM) staging). The pathological TNM stage was determined according to the 7th American Joint Committee on Cancer (AJCC) staging system. Information of complete blood count, differential count, biochemistry profiles, RT and cisplatin doses were also collected from medical records.

### 2.1. Radiotherapy

All patients received three-dimensional conformal or intensity modulated RT on 5 consecutive days per week at a conventional fractionated daily dose of 1.8–2.0 Gray. For each patient, a non-contrasted computed tomography scan with a 3.75 mm slice thickness was performed in the treatment position with an immobilization mask. Gross tumor volume was defined as the tumor area and involved lymphadenopathy detected on computed tomography (CT) scan, magnetic resonance imaging (MRI) or positron emission tomography-computed tomography. RT is administered with high-energy photons beams of 6–10 MV to a total planned dose of 60–74 Gray in 30 to 37 fractions over 6–8 weeks to gross tumor volume.

### 2.2. Chemotherapy

According to the clinical practice guideline for multidisciplinary esophageal team in Changhua Christian Hospital, the systemic CHT regimens of CCRT included triweekly cisplatin plus 5-fluorouracil (PF). The choice of regimen was based on patient clinical presentation and attending physician experience. The triweekly PF regimen consisted of cisplatin 75 mg/m$^2$ as a 4 h intravenous infusion and fluorouracil 1000 mg/m$^2$ per 24 h as a 96 h continuous infusion, repeated every 3 weeks or 4 weeks.

### 2.3. Hematological Profile before and after CCRT

Complete blood count and differential count were obtained before and after first cycle of CT. Prechemotherapy hemoglobin level (Hb0), platelet count (PLT0), neutrophil-to-

lymphocyte ratio (NLR0), and platelet-to lymphocyte ratio (PLR0) were obtained within one week before CHT. Postchemotherapy hemoglobin level (Hb1), platelet count (PLT1), neutrophil-to-lymphocyte ratio (NLR1), and platelet-to lymphocyte ratio (PLR1) were obtained within one week before second cycle of CHT.

*2.4. Outcome Analysis and Adverse Events*

Image modalities, such as CT or MRI, were performed to confirm or exclude disease progression after complete CCRT. Disease-specific survival (DSS) was defined as the time elapsed between the start of initial diagnosis and the date of death due to cancer, or if the patient was still alive 5 years after the start of diagnosis. DSS was calculated from the date of the cancer diagnosis to the date of cancer-related death. Overall survival (OS) was defined as the time elapsed between the start of initial diagnosis and the date of death of any cause, or if the patient was still alive 5 years after the start of diagnosis. OS was calculated from the date of cancer diagnosis to the date of death of any cause. Patients who were lost to follow-up within 5 years were censored at their last date of follow-up. In the analysis of DSS, deaths due to causes other than esophageal cancer were treated as censored observations at the time of death.

*2.5. Statistical Analysis*

Chi-square test was used for comparison of categorical variables. Independent sample t-test was used to compare continuous variables. The Kaplan–Meier method was applied to estimate OS and DSS for various group partitions. Log-rank test was carried out to compare survival rate between various groups with different risk factors. Cox proportional hazard models were used to assess the effects of potential risk factors. Univariate and multivariate Cox regression analyses were used to assess the association between survival and patient, disease, and treatment characteristics. Optimal cutoff of each factor was calculated by Youden's J statistic. Variables with a *p*-value of 0.2 or less on univariate analysis were selected to enter a backward selection algorithm to yield the parsimonious multivariable regression model. The assumption for proportional hazards was evaluated by using scaled Schoenfeld residuals. Hazard ratios and 95% confidence intervals (CI) were shown. Statistical analysis was performed by using the SPSS software 22nd version (SPSS Inc., Chicago, IL, USA) and R software (version 4.0.3). Two-tailed *p* < 0.05 was considered statistically significant.

**3. Results**

*3.1. Patient and Tumor Characteristics*

A total of 420 Esophageal SCC patients were retrospectively included from January 2008 to December 2017. The demographic and clinical characteristics are provided in Table 1. There were 397 men and 23women with a median age of 55 years (range 33–87); 80.2% of the patients were <65 years old. Three hundred and eighty five (91.7%) patients had an ECOG performance status of 0–1, and 35 (8.3%) patients had a performance status of 2–3. The primary site of tumor was at upper third esophagus in 122 patients (29.0%), middle third esophagus in 122 patients (42.4%), and lower third esophagus in 120 patients (28.6%). Two hundred and eighty two patients (67.1%) were stage III/IV, whereas 138 (32.9%) patients were stage I/II. One hundred and fifty (35.7%) patients underwent surgical resection after CCRT.

**Table 1.** Demographic and clinical characteristics of 420 enrolled patients.

| Variables. | | N (%) or Mean ± SD |
|---|---|---|
| Age (years) | <65 | 337 (80.20) |
| | ≥65 | 83 (19.80) |
| Gender | Male | 397 (94.50) |
| | Female | 23 (5.50) |
| ECOG performance status | 0–1 | 385 (91.7%) |
| | 2–3 | 35 (8.3%) |
| Primary site | Upper 1/3 | 122 (29.0) |
| | Middle 1/3 | 178 (42.4) |
| | Lower 1/3 | 120 (28.6) |
| Staging | I | 20 (4.8) |
| | II | 118 (28.1) |
| | III | 223 (53.1) |
| | IV | 59 (14.0) |
| T stage | T0-1 | 25 (6.0) |
| | T2-4 | 393 (94.0) |
| N stage | N0-1 | 232 (55.5) |
| | N2-3 | 186 (44.5) |
| M stage | 0 | 361 (86.0) |
| | 1 | 59 (14.0) |
| Differentiation | well | 5 (1.2) |
| | Moderate | 321 (76.4) |
| | Poor | 48 (11.4) |
| | unkown | 46 (11.0) |
| Operation | No | 270 (64.30) |
| | Yes | 150 (35.70) |
| RT dose (cGY) | ≥5000 | 376 (89.5) |
| | <5000 | 44 (10.5) |
| Cisplatin dose (mg/m$^2$) | ≥150 | 336 (80) |
| | <150 | 84 (20) |
| WBC0 (K/ mm$^3$) | | 7.8 ± 3.01 |
| Hb0 (g/dL) | | 12.4 ± 1.98 |
| PLT0 (10$^9$/L) | | 258 ± 98.71 |
| WBC1 (K/ mm$^3$) | | 5.4 ± 3.5 |
| Hb1 (g/dL) | | 11.9 ± 1.88 |
| PLT1 (10$^9$/L) | | 199 ± 94.8 |
| NLR0 | ≥3.5 | 276 (65.7) |
| | <3.5 | 144 (34.3) |
| PLR0 | ≥375 | 91 (21.7) |
| | <375 | 329 (78.3) |
| NLR1 | ≥6.9 | 220 (52.4) |
| | <6.9 | 200 (47.6) |
| PLR1 | ≥463 | 187 (44.5) |
| | <463 | 233 (55.5) |
| NLR1/NLR0 | ≥1 | 289 (68.8) |
| | <1 | 131 (31.2) |
| PLR1/PLR0 | ≥1 | 323 (76.9) |
| | <1 | 97 (23.1) |

WBC0: white blood cell count before CHT; Hb0: hemoglobin level before CHT; PLT0: platelet count before CHT; NLR0: neutrophil-to-lymphocyte ratio before CHT; PLR0: platelets-to-lymphocyte ratio before CHT; WBC1: white blood cell count after first cycle of CHT; Hb1: hemoglobin level after first cycle of CHT; PLT1:platelet count after first cycle of CHT; NLR1: Neutrophil-to-lymphocyte ratio after first cycle of CHT; PLR1: Platelets-to-lymphocyte ratio after first cycle of CHT.

### 3.2. Univariate and Multivariate Cox Regression Analysis for OS and DSS

On univariate analysis for OS, variables with a *p*-value of 0.2 or less were allowed to enter a backward selection algorithm to yield the parsimonious multivariable regression model (Table 2). Significant factors for OS on univariate analysis included ECOG performance status, clinical staging, operation, RT dose, cisplatin dose, Hb0, NLR0, PLR0, Hb1, PLT1, and NLR1/NLR0. On univariate analysis for DSS, significant factors included ECOG performance status, clinical staging, operation, cisplatin dose, NLR0, PLR0, PLR1/PLR0, and NLR1/NLR0. Multivariate Cox regression results showed that ECOG performance status 0–I (*p* = 0.031, HR (95%CI) = 1.498 (1.039–2.162)), Clinical staging I–II (*p* < 0.001, HR (95%CI) = 1.66 (1.303–2.116)), Operation (*p* < 0.001, HR (95%CI) = 0.473 (0.372–0.601)), cisplatin dose > 150 mg/m$^2$ (*p* < 0.001, HR (95%CI) = 0.565 (0.431–0.741)), PLR0 < 375 (*p* = 0.004, HR (95%CI) = 1.463 (1.13–1.895)), and PLT1 > 150 × 10$^9$/L (*p* = 0.011, HR (95%CI) = 0.74 (0.588–0.932)) were independent factors for predicting OS.

**Table 2.** Univariate Analysis and Multivariate Cox proportional Hazard Regression Analysis for OS and DSS.

| | Univariate Analysis | | | | Multivariate Analysis | | | |
| --- | --- | --- | --- | --- | --- | --- | --- | --- |
| | OS | | DSS | | OS | | DSS | |
| Characteristics | cHR (95%CI) | *p* Value | cHR (95%CI) | *p* Value | aHR (95%CI) | *p* Value | aHR (95%CI) | *p* Value |
| Age (years) | | | | | | | | |
| ≥65 vs. <65 | 1.148 (0.881–1.496) | 0.308 | 1.019 (0.742–1.399) | 0.909 | | | | |
| Gender | | | | | | | | |
| female vs. male | 0.795 (0.481–1.314) | 0.372 | 0.792 (0.443–1.414) | 0.43 | | | | |
| ECOG performance status | | | | | | | | |
| 3–4 vs. 0–1 | 1.702 (1.192–2.43) | 0.003 | 1.68 (1.11–2.543) | 0.014 | 1.498 (1.039–2.162) | 0.031 | 1.566 (1.025–2.392) | 0.038 |
| Clinical staging | | | | | | | | |
| III–IV vs. I–II | 1.709 (1.347–2.168) | <0.001 | 2.024 (1.524–2.69) | <0.001 | 1.66 (1.303–2.116) | <0.001 | 1.988 (1.49–2.651) | <0.001 |
| Operation | | | | | | | | |
| yes vs. no | 0.458 (0.362–0.58) | <0.001 | 0.457 (0.349–0.6) | <0.001 | 0.473 (0.372–0.601) | <0.001 | 0.473 (0.359–0.622) | <0.001 |
| RT dose (cGY) | | | | | | | | |
| >5000 vs. <5000 | 0.631 (0.453–0.878) | 0.006 | 0.738 (0.491–1.108) | 0.143 | | | | |
| Cisplatin dose (mg/m$^2$) | | | | | | | | |
| >150 vs. <150 | 0.576 (0.443–0.748) | <0.001 | 0.558 (0.413–0.753) | <0.001 | 0.565 (0.431–0.741) | <0.001 | 0.515 (0.378–0.7) | <0.001 |
| Hb0 (g/dL) | | | | | | | | |
| ≥10 vs. <10 | 0.648 (0.473–0.888) | 0.007 | 0.766 (0.52–1.13) | 0.179 | | | | |
| PLT0 (10$^9$/L) | | | | | | | | |
| ≥150 vs. <150 | 0.801 (0.576–1.115) | 0.189 | 0.984 (0.651–1.487) | 0.937 | | | | |
| NLR0 | | | | | | | | |
| ≥3.5 vs. <3.5 | 1.427 (1.132–1.798) | 0.003 | 1.513 (1.156–1.98) | 0.003 | | | | |
| PLR0 | | | | | | | | |
| ≥375 vs. <375 | 1.4 (1.085–1.808) | 0.01 | 1.475 (1.103–1.973) | 0.009 | 1.463 (1.13–1.895) | 0.004 | 1.532 (1.143–2.054) | 0.005 |
| Hb1 (g/dL) | | | | | | | | |
| ≥10 vs. <10 | 0.648 (0.489–0.857) | 0.002 | 0.74 (0.527–1.039) | 0.082 | | | | |
| PLT1 (10$^9$/L) | | | | | | | | |

**Table 2.** *Cont.*

| | Univariate Analysis | | | | Multivariate Analysis | | | |
| | OS | | DSS | | OS | | DSS | |
| Characteristics | cHR (95%CI) | *p* Value | cHR (95%CI) | *p* Value | aHR (95%CI) | *p* Value | aHR (95%CI) | *p* Value |
|---|---|---|---|---|---|---|---|---|
| ≥150 vs. <150 | 0.788 (0.629–0.988) | 0.039 | 0.909 (0.697–1.187) | 0.484 | 0.74 (0.588–0.932) | 0.011 | | |
| NLR1 | | | | | | | | |
| ≥6.9 vs. <6.9 | 1.025 (0.827-1.27) | 0.819 | 1.013 (0.792-1.298) | 0.916 | | | | |
| PLR1 | | | | | | | | |
| ≥463 vs. <463 | 1.043 (0.841-1.294) | 0.699 | 1.013 (0.79-1.299) | 0.917 | | | | |
| PLR1/PLR0 | | | | | | | | |
| ≥1 vs. <1 | 0.798 (0.622–1.024) | 0.076 | 0.724 (0.546–0.959) | 0.024 | | | | |
| NLR1/ NLR0 | | | | | | | | |
| ≥1 vs. <1 | 0.777 (0.619–0.975) | 0.029 | 0.733 (0.565–0.95) | 0.019 | | | | |

cHR: crude hazard ratio; aHR: adjusted harzard ratio; CI: confidence interval; RT: radiotherapy; CHT: chemotherapy; Hb0:hemoglobin level before CHT; PLT0: platelet count before CHT; NLR0: neutrophil-to-lymphocyte ratio before CHT; PLR0: platelets-to-lymphocyte ratio before CHT; Hb1: hemoglobin level after first cycle of CHT; PLT1:platelet count after first cycle of CHT; NLR1: Neutrophil-to-lymphocyte ratio after first cycle of CHT; PLR1: Platelets-to-lymphocyte ratio after first cycle of CHT.

Independent factors for predicting DSS included ECOG performance status 0–I ($p = 0.038$, HR (95%CI) = 1.566 (1.025–2.392)), Clinical staging I–II ($p < 0.001$, HR(95%CI) = 1.988 (1.49–2.651)), Operation ($p < 0.001$, HR (95%CI) = 0.473 (0.359–0.622)), cisplatin dose > 150 mg/m$^2$ ($p < 0.001$, HR (95%CI) = 0.515 (0.378–0.7)), and PLR0 < 375 ($p = 0.005$, HR (95%CI) = 1.532 (1.143–2.054)).

### 3.3. Survival Analysis According to PLR and NLR

There was a significant difference in survival between patients with high PLR0 and low PLR0. Patients with PLR0 < 375 had a median OS of 15 months, compared with 11 months for those with PLR0 ≥ 375 ($p = 0.008$ by the log rank test, Figure 1a). Patients with PLR0 < 375 had a median DSS of 19 months, compared with 14 months for those with PLR0 ≥ 375 ($p = 0.007$, Figure 1b). There was a significant difference in survival between patients with high NLR0 and low NLR0. Patients with NLR0 < 3.5 had a median OS of 17 months, compared with 13 months for those with NLR0 ≥ 3.5 ($p = 0.002$ by the log rank test, Figure 2a). Patients with NLR0 < 3.5 had a median DSS of 31 months, compared with 15months for those with NLR0 ≥ 3.5 ($p = 0.002$, Figure 2b). There was no significant difference in OS between patients with PLR1/PLR0 ≥ 1 and PLR1/PLR0 < 1. Patients with PLR1/PLR0 ≥ 1 had a median OS of 14 months, compared with a 12 months for those with PLR1/PLR0 < 1 ($p = 0.069$ by the log rank test, Figure 3a). However, there was a significant difference in DSS between patients with PLR1/PLR0 ≥ 1 and PLR1/PLR0 < 1. Patients with PLR1/PLR0 ≥ 1 had a median DSS of 20 months, compared with a 15 months for those with PLR1/PLR0 < 1 ($p = 0.021$, Figure 3b). There was a significant difference in survival between patients with NLR1/NLR0 ≥ 1 and NLR1/NLR0 < 1. Patients with NLR1/NLR0 ≥ 1 had a median OS of 14 months, compared with a 13 months for those with NLR1/NLR0 < 1 ($p = 0.025$, Figure 4a). Patients with NLR1/NLR0 ≥ 1 had a median DSS of 20 months, compared with a 15 months for those with NLR1/NLR0 < 1 ($p = 0.016$, Figure 4b).

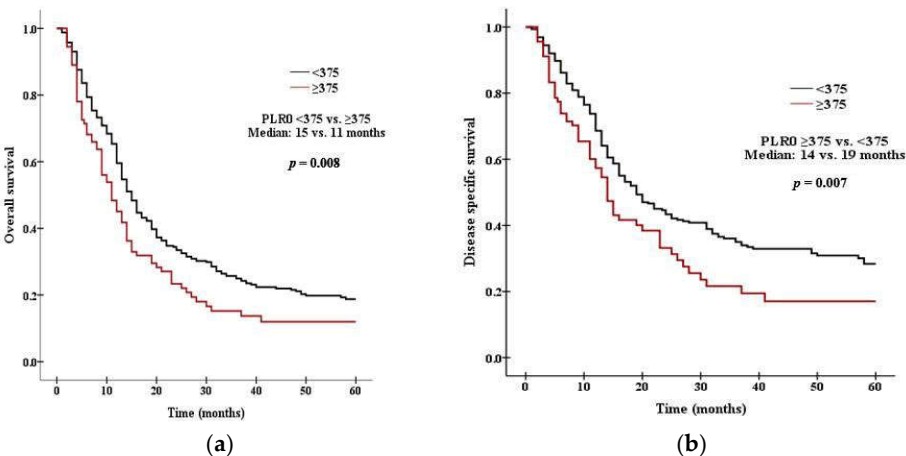

**Figure 1.** Kaplan–Meier curves of (**a**) OS in patients with PLR0 < 375 versus PLR0 ≥ 375, (**b**) DSS in patients with PLR0 < 375 versus PLR0 ≥ 375.

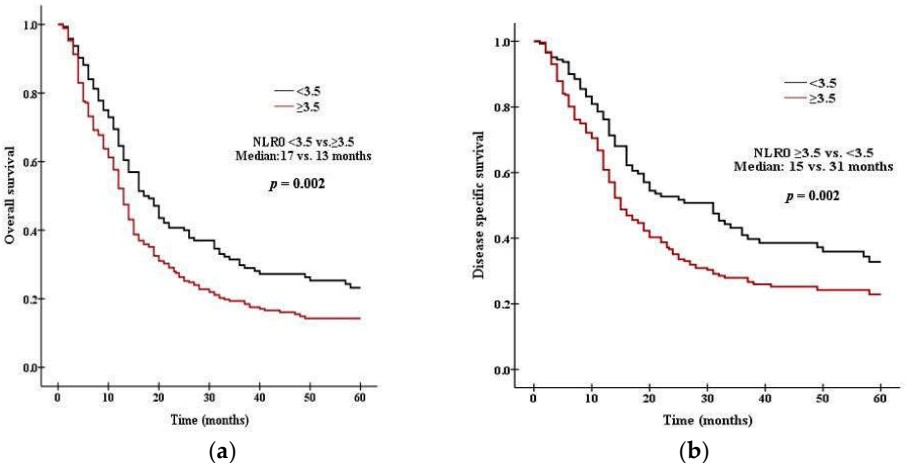

**Figure 2.** Kaplan–Meier curves of (**a**) OS in patients with NLR0 < 3.5 versus NLR0 ≥ 3.5, (**b**) DSS in patients with NLR0 < 3.5 versus NLR0 ≥ 3.5.

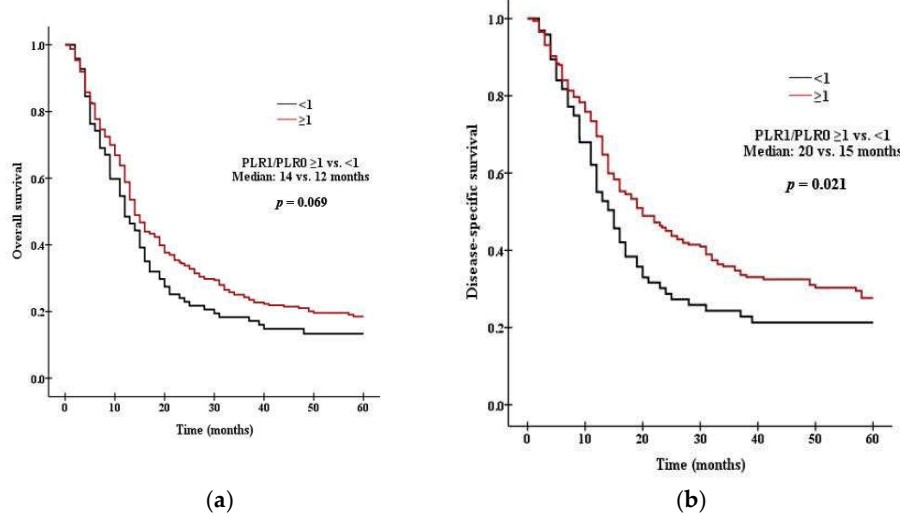

**Figure 3.** Kaplan–Meier curves of (**a**) OS in patients with PLR1/PLR0 ≥ 1 versus PLR1/PLR0 < 1, (**b**) DSS in patients with PLR1/PLR0 ≥ 1 versus PLR1/PLR0 < 1.

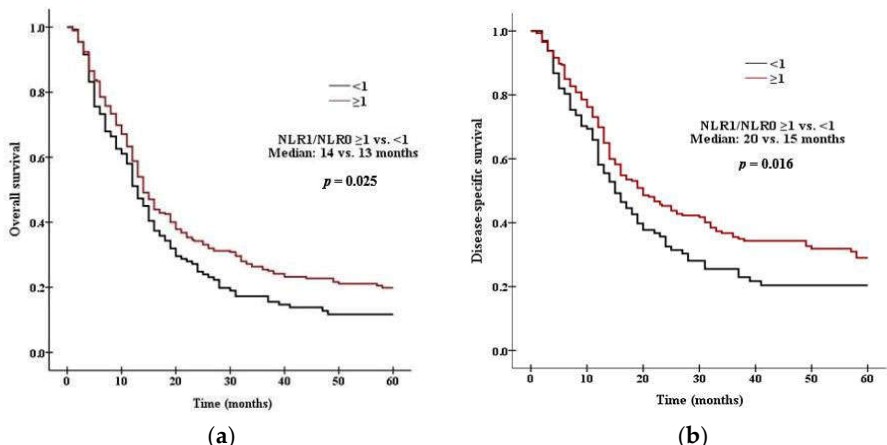

**Figure 4.** Kaplan–Meier curves of (**a**) OS in patients with NLR1/NLR0 $\geq$ 1 versus NLR1/NLR0 < 1, (**b**) DSS in patients with NLR1/NLR0 $\geq$ 1 versus NLR1/NLR0 < 1.

## 4. Discussion

Definitive CCRT is an established management option for patients with locally advanced esophageal cancer and the overall survival outcome of CCRT is comparable to that of preoperative CCRT plus surgery [4–6]. Some clinical factors have been used to predict outcome of esophageal cancer patients receiving CCRT. In our analysis of 420 patients with esophageal SCC who underwent CCRT, advanced age of $\geq$65 years old was not a significant factor for poor OS ($p$ = 0.308) and DSS ($p$ = 0.909). The result was compatible with a systemic review of outcome of elderly esophageal cancer patients. In this article, advanced age seems not to be an influencing factor for treatment outcome, although 22–36% patients had CCRT toxicity grade of more than 3 [11].In our study, it was shown that ECOG performance status 1–2, clinical staging I–II, operation, and cisplatin dose >150 mg/m$^2$ were significantly associated with OS and DSS; furthermore, these four factors were independent predictors for OS and DSS on multivariate analysis (Table 2). Similar results were observed in a recent study of esophageal cancer patients undergoing CCRT [6]. The authors concluded that poor performance status, advanced stage, poor CCRT response were independent predictors of poor survival.

Anemia has been thought to be an influencing factor of treatment outcome for patients receiving CCRT. In a study by Duron JJ et al., they demonstrated that anemia is an independent risk factor for increased mortality during treatment of esophageal cancer patients [12]. Melis M et al., showed that neoadjuvant therapy to esophageal cancer patients increased rate of perioperative anemia and overall complications [13]. Prechemotheray Hb0 < 10 g/dL was significantly associated with poor OS ($p$ = 0.007, Table 2) in our study, and the postchemotherapy Hb1 < 10 g/dL was also a significant factor for poor OS. Our study confirmed that prechemotherapy anemia and anemia after first cycle of CCRT were both significant factors for poor survival outcome. The prechemotherapy platelet count was not significantly associated with OS and DSS in our study. However, it is worth noting that postchemotherapy platelet count $\geq$150 $\times$ 10$^9$/L was significantly associated with better OS, and was shown to be an independent predictor for OS on multivariate analysis ($p$ = 0.011, Table 2). Our results were different from those of pervious articles using platelet count as a prognostic markers [14–16]. The cutoff values of platelet count used in these 3 articles were among 293 to 300 $\times$ 10$^9$/L; however, the optimal cutoff used in our study was calculated using Youden index, thus causing the different result of our study from previous studies.

Elevated PLR was considered to be a poor prognostic marker for esophageal cancer patients in several studies. One recent meta-analysis showed that elevate PLR can predict poor OS (HR (95%CI) = 1.389 (1.161–1.663), DSS (HR (95%CI) = 1.686 (1.146–2.480)), and disease-free survival (HR (95%CI) = 1.404 (1.169–1.687)) for esophageal cancer patients [7]. In our study, we performed univariate and multivariate analysis and found that PLR0 $\geq$ 375 was significantly associated with poor OS and DSS. PLR0 $\geq$ 375 was further shown to be

an independent predictor for poor OS on multivariate analysis. Moreover, interestingly, PLR1/PLR0 $\geq$ 1 was significantly associated with better DSS ($p$ = 0.024), suggesting that patients with elevated PLR after first cycle of CHT had better DSS than those with decreased PLR. This finding of our study is similar to a previous study on the change of NLR and PLR at various time intervals. In this retrospective study of nonmetastatic esophageal cancer patients, Hyder J et al., showed that a better progression free survival was noted in patients with higher postchemotherapy PLR [17]. Elevated NLR was also consider to be prognostic marker for esophageal cancer patients. The pooled results of a meta-analysis showed that elevated NLR might predict poor OS, DSS, progression free survival and disease free survival [8]. Our study showed that NLR0 $\geq$ 3.5 was significantly associated with poor OS and DSS. Our study further showed that NLR1/NLR0 $\geq$1 was significantly associated with better OS and DSS, suggesting that patients with elevated NLR after first cycle of CHT had better OS and DSS than those with decreased NLR. Hyder J et al., also demonstrated that a higher probability of complete pathological response was noted in patients with higher postchemotherapy NLR [17]. Radiotherapy may reduce peripheral blood cell counts and lymphocytes are more radiosensitive than other leukocytes [18]. Chemotherapy adversely affects the hematopoietic system, and neutropenia and thrombocytopenia are the most serious hematologic toxicities of CHT [19,20]. Because the neutrophil and platelet count in our study were checked within one week before second cycle of CHT, the observed better survival in patients with elevated postchemotherapy PLR and NLR might indicate a better recovery from chemotherapy-induced myelosuppression in this subgroup of patients.

There were some limitations in our study. Firstly, this study was conducted in a retrospective fashion and all included patients were from a single institution. The treatment decisions were based only on evaluations by the primary physician and multidisciplinary team and might have some selection bias. Secondly, some risk factors were not included in this study, e.g., number of comorbidity and clinical response to treatment, and might have influence on the result of prediction model.

In conclusion, our study showed that ECOG performance status, clinical staging, operation, cisplatin dose, prechemotherapy PLR, and postchemotherapy platelet count were independent predictors of OS for esophageal SCC patients receiving CCRT. ECOG performance status, clinical staging, operation, cisplatin dose, and prechemotherapy PLR were independent predictors of DSS. Low levels of prechemotherapy PLR and NLR were associated with better OS and DSS. Elevated postchemotherapy platelet count and NLR were associated with better OS. Elevated postchemotherapy PLR and NLR were associated with better DSS. The finding that patients with elevated postchemotherapy NLR and PLR had better survival and might suggest a better recovery from chemotherapy-induced myelosuppression in this subgroup of patients. Because our study is conducted in a retrospective fashion, further prospective studies are needed to elucidate the findings of our study.

**Author Contributions:** Data curation, R.-H.T. and K.-M.L.; Investigation, R.-H.T. and K.-M.L.; Software, C.-Y.T.; Methodology, Conceptualization and Supervision, S.-L.Y. All authors have read and agreed to the published version of the manuscript.

**Funding:** This research received no external funding.

**Institutional Review Board Statement:** The study was conducted in accordance with the Declaration of Helsinki, and approved by the institutional review board of Changhua Christian Hospital (IRB No. 190123).

**Informed Consent Statement:** Patient consent was waived due to a retrospective study.

**Data Availability Statement:** No new data were created or analyzed in this study. Data sharing is not applicable to this article.

**Conflicts of Interest:** The authors declare no conflict of interest.

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
