# Peer review of "Elevated Platelet-to-Lymphocyte Ratio and Neutrophil-to-Lymphocyte Ratio after First Cycle of Chemotherapy and Better Survival in Esophageal Cancer Patients Receiving Concurrent Chemoradiotherapy"

_curroncol, doi:10.3390/curroncol29110694_

Round 1

Reviewer 1 Report

The manuscript reports the study of platelet-to-lymphocyte ratio (PLR) and neutrophil-to-lymphocyte ratio (NLR) in patients with esophageal cancer who underwent concurrent chemoradiotherapy (CCRT). The study is interesting, the number of patients and statistical methods seem to be enough. My comments are as follow.

Major Comments

#1: There are many reports of PLR and NLR in patients with esophageal cancer. As a result, many systematic review and meta-analysis are found.

For example,

 - the reference 7 (meta-analysis).

 - the reference 8 (meta-analysis).

 - http://dx.doi.org/10.21037/jtd.2019.07.30 (systematic review and meta-analysis).

 - DOI 10.1245/s10434-015-4869-5 (systematic review and meta-analysis).

Therefore, I don’t think the retrospective study has novelty. To higher the novelty of the study, a rigid literature review must be written, and state “what is known, and what is unknown”.

#2: page 2 (Methods section):

Please clarify that informed consent was obtained from the candidates for the study, or not. Opt-out method?

Minor Comments

#3: page 1:

The references 1―5 is too old. Newer ones are preferred.

#4: page 1:

Please replace all “CT” with “CHT”.

#5: page 2:

The below sentence might be deleted because it is not about esophageal squamous cell carcinoma.

“Another study by Tankel J et al. showed that perioperative change in PLR was associated with poor overall survival in patients with esophageal adenocarcinoma [10].”

#6: page 2 (Methods section):

Please clarify how to determine the clinical stage of the patient. With the Union Against Cancer 8th edition?

Also, why were the patients with T4 stage excluded from the study candidates? Please clarify.

#7: pages 9―10 (References section):

Please arrange all the references according to the Journal’s guidelines.

Author Response

Response to Reviewer 1 

Major Comments

#1: There are many reports of PLR and NLR in patients with esophageal cancer. As a result, many systematic review and meta-analysis are found.

For example,

 - the reference 7 (meta-analysis).

 - the reference 8 (meta-analysis).

 - http://dx.doi.org/10.21037/jtd.2019.07.30 (systematic review and meta-analysis).

 - DOI 10.1245/s10434-015-4869-5 (systematic review and meta-analysis).

Therefore, I don’t think the retrospective study has novelty. To higher the novelty of the study, a rigid literature review must be written, and state “what is known, and what is unknown”.

Response:

A paragraph discussing these four meta-analyses is added in page 2 and marked red.

#2: page 2 (Methods section):

Please clarify that informed consent was obtained from the candidates for the study, or not. Opt-out method?

Response:

Because our study was a retrospective study, the IRB approved a waiver for informed consent. We add a sentence” The IRB approved a waiver for informed consent.” in page 2.

Minor Comments

#3: page 1:

The references 1―5 is too old. Newer ones are preferred.

Response:

We have replaced references 2-5 with recent publications.

#4: page 1:

Please replace all “CT” with “CHT”.

Response:

We have replaced all “CT” with “CHT”, and all changes are marked red.

#5: page 2:

The below sentence might be deleted because it is not about esophageal squamous cell carcinoma.

“Another study by Tankel J et al. showed that perioperative change in PLR was associated with poor overall survival in patients with esophageal adenocarcinoma [10].”

 Response:

The sentence is deleted and marked blue and underlined in page 2.

#6: page 2 (Methods section):

Please clarify how to determine the clinical stage of the patient. With the Union Against Cancer 8th edition?

Also, why were the patients with T4 stage excluded from the study candidates? Please clarify.

Response:

  1. The clinical stages of all included patients were determined according to AJCC 7th edition.
  2. Patients with T4 stage were not excluded. The “T2-3” in page 4 was mistyped and is corrected as” T2-4” (marked red).

#7: pages 9―10 (References section):

Please arrange all the references according to the Journal’s guidelines.

Response:

All references are corrected according to journal’s guideline.

Reviewer 2 Report

It was a pleasure reading the manuscript entitled, “Elevated platelet-to-lymphocyte ratio and neutrophil-to-lymphocyte ratio after first cycle of chemotherapy and better survival in esophageal cancer patients receiving concurrent chemoradiotherapy”. This is a single institution retrospective study of patients with esophageal SCC who have undergone neoadjuvant chemoradiation.

The design of the study was well explained and straightforward. The results are not surprising and the authors put them into context appropriately. Aside from flow and grammar, this is an excellent study.

Minor comment:

In the calculation of DSS, the authors mention if the patient was still alive 5 years after the start of the diagnosis. I think they mean that they only counted up to 5 years post diagnosis, but they should clarify.

Author Response

Response to Reviewer 2

Minor comment:

In the calculation of DSS, the authors mention if the patient was still alive 5 years after the start of the diagnosis. I think they mean that they only counted up to 5 years post diagnosis, but they should clarify.

Response:

To avoid confusion, we have deleted “or if the patient was still alive 5 years after the start of diagnosis” in second and third sentences in page 3 (marked blue and underlined)

Round 2

Reviewer 1 Report

Thanks for the revisions. The revisions are sufficient.